# An Optimal Beneficiary Profile to Ensure Focused Interventions for Older Adults

**DOI:** 10.3390/geriatrics10020059

**Published:** 2025-04-14

**Authors:** Dorina-Claudia Bălan, Rozeta Drăghici, Ioana Găiculescu, Alexandra Rusu, Andrada-Elena Stan, Polixenia Stan

**Affiliations:** Research Laboratory of Social Gerontology and GerontoPsychology, “Ana Aslan” National Institute of Gerontology and Geriatrics, 011241 Bucharest, Romania; claudiabalan@ana-aslan.ro (D.-C.B.); ioanagaiculescu@ana-aslan.ro (I.G.); alexandrarusu@ana-aslan.ro (A.R.); andrada.stan@ana-aslan.ro (A.-E.S.); polixenia.stan@ana-aslan.ro (P.S.)

**Keywords:** optimal beneficiary profile, yoga classes, older people, quality of life, mixed methods

## Abstract

**Background:** Aging is a lifelong process, and many chronic diseases and geriatric syndromes are influenced by lifestyle factors. For active aging and maintaining functional capacity facilitate health, there are essential aspects in geriatric care. Our objective was to create a specific profile focusing on the characteristics of a possible optimal beneficiary of a newly developed program that is meant to increase the social inclusion and participation in social life of older adults. **Methods:** The profile was built based on a mixed design, a quantitative and qualitative analysis that identified the typology of an optimal beneficiary of a newly developed yoga program. The quantitative analysis (50 subjects from NIGG “Ana Aslan” Bucharest) identified the main predictors impacting subjects’ willingness to participate in a yoga program based on their pathologies at a mental and/or physical level. The main materials used for this were the Clinical Assessment Scales for the Elderly (CASE-SF) and the Quality-of-Life Assessment Questionnaire (WHOQOL-BREF). The qualitative analysis consisted of four focus groups (10 subjects from NIGG “Ana Aslan” and 7 subjects from GNSPY), aiming to provide the in-depth reasons for participating in a yoga program. **Results:** The results showed that a diagnosed physical impairment was correlated with an increased willingness to participate in yoga classes, while a mental pathology was associated with a decreased willingness to participate in such a program. Five main themes emerged from the qualitative analysis. **Conclusions:** The profile provides answers related to the specifics of the beneficiary based on their motivation, limits, and personality traits.

## 1. Introduction

Demographic change is an important and current issue that concerns all decision-makers at both international and national levels. In the field of health, population aging has come to the attention of practitioners from the point of view of intervention strategies to ensure an optimal quality of life and from the perspective of the costs involved. The rate of this process is “unparalleled in human history” (United Nations, Department of Economic and Social Affairs, Population Division, 2001: xxviii) [1]. By 2030, one in six people in the world will be aged 60 years or over. At this time, the share of the population aged 60 years and over will increase from 1 billion in 2020 to 1.4 billion. By 2050, the world’s population of people aged 60 years and older will double (2.1 billion). The number of people aged 80 years or older is expected to triple between 2020 and 2050, reaching 426 million [2].

In Romania, based on the demographic pyramid model, in 2020, we are talking about a narrow base that starts to grow starting from the age of 40 [3]. Eurostat data estimate that the elderly resident population (aged 65 and over) of Romania will reach 3.82 million by 2030 and, in 2060, the population aged 65 and over will reach 4.72 million, compared to 3.71 million people on 1 January 2022 [4].

The general trend of population aging involves a series of challenges related to the associated physical and mental health. At the biological level, aging results in a gradual decline in physical and mental capacities and an increased risk of disease. Furthermore, as people age, they are more likely to experience multiple diseases simultaneously and a single disease can become a risk factor for several medical conditions. Hearing loss [5,6], cataracts [7], back and neck pain, osteoarthritis [8], osteoporosis [9], chronic obstructive pulmonary disease [10], diabetes [11], depression, and dementia [12] are common problems in older adults. Aging is a lifelong process, and older adults are more susceptible to lifestyle-related diseases, such as stroke, coronary artery disease [13], and cancer [14,15]. Also, many geriatric syndromes are influenced by lifestyle factors [16]. In addition to biological changes, retirement, loss of purpose in life, relocation to more suitable housing, and the death of a friend or partner often result in psychological damage and affect the quality of life of long-lived older people. Factors that are beneficial to active aging and maintaining a functional capacity to facilitate health and peaceful living during the last part of life are essential in geriatric care.

Because the polypharmacy approach has its own limits and increases risks of adverse reactions, mind–body medicine, such as like yoga and meditation, could be a way to achieve balance between mind, body, and spirit. Yoga is a product of the civilization of India and it refers to an art of living at the highest level in connection with the larger reality of life [17]. Yoga practices are known for the positive effects they have on various parameters of psychological functioning. Yoga, a physical and mental activity starting from the body, has benefits for mental, physical, and spiritual health. There are nine primary styles commonly practiced in the Western world, but yoga could encompass a wide range of practices. Yoga is a diverse and evolving practice with numerous schools, traditions, and styles. Most contemporary practice is based on Hatha yoga that is largely defined by physical practice [18]. Each type is suitable depending on various factors like physical condition, personal preferences, and specific health objectives. Recent interventions have adapted these types for a variety of health conditions problems in aging [19], chronic pain [20], diabetes [21], stroke [22], and heart failure.

A three-month-long complex yoga-based intervention conducted on a Hungarian community sample of 73 participants with an age range of between 30 and 64 revealed the positive impact on many aspects of healthy psychological functioning. The intervention reduced participants’ perceived stress and negative affectivity, improved spirituality, and various aspects of interoceptive awareness, but it did not impact positive affectivity and life satisfaction [23]. Yoga practices can be adapted for the specific needs of older adults. Hatha yoga includes different interventions for increased balance and mobility in people 60 years and older [24], whereas Gentle Years Yoga for older adults uses a mixture of standing, seated, kneeling, supine, and prone stationary positions, altering Hatha Yoga poses to make them accessible and safe for inactive older adults with comorbidities and physical and balance limitations [25].

Emotions are integral to the human experience, shaping our perceptions, decisions, and overall wellbeing. A cognitive process known as interoception plays a crucial role in interpreting and regulating our emotions—it enables us to tune into our internal states, decode signals, and ultimately take charge of our wellbeing. Interoception, or the capacity of self-awareness, refers to the ability to sense and interpret internal bodily signals, including sensations from our organs, muscles, and skin. It involves perceiving and integrating information from within our bodies, creating a comprehensive representation of our physiological state. Interception tells us when we are tired, stressed, or happy. Evidence is built from different disciplines with regards to impairment of interoceptive ability and mental health conditions suggesting that interoceptive ability is “a transdiagnostic process and is impaired in emotional disorders” [26] and “conversely associated with good mental health and increased empathy and emotional regulation” [27]. Self-awareness is not a fixed trait but a skill that can be developed and refined over time. Practices such as meditation, body scans, and journaling can enhance interoception abilities by fostering present-moment awareness and deepening the mind–body connection. Yoga, as science and an art of a conscious life, involves a special focus on these aspects. This is because increased self-awareness leads to increased self-care, self-compassion, curiosity, openness to other-directed relationships, and interactions through increased participation in learning activities and in social life. Evidence also comes from a complex meta-analysis conducted by Eurich (2018), showing that self-awareness, as a key human competence, is associated with increased relationship satisfaction as well as elevated social and personal control and happiness, while having a negative relationship with anxiety, stress, and depression [28]. Yoga is a practice that can promote this ability by focusing our attention on bodily sensations. As a result, there has been an increased interest in interventions that focus on our interoceptive ability through yoga interventions.

The “Increasing the quality of life of older people with fewer opportunities in Bucharest and Rome” Erasmus+ project aims to develop the competence of self-awareness of one’s body, breathing, emotions, feelings, and thoughts to increase social inclusion and participation in social life through a yoga program adapted to the needs of older adults. Our main motivation for developing this project is the analysis of the situation of the older adults whom we consider Adult Learners. The aim of the project is to increase one’s competence to be more aware of their body (i.e., sensations, joints, different body parts, and movements), of their breathing, emotions, feelings, and mind.

Developing an effective yoga intervention program that meets the needs of older adults was necessary to have as much in-depth knowledge as possible of the beneficiary typology that would benefit the most from this program. Thus, the idea of creating an optimal beneficiary profile emerged. An optimal beneficiary profile refers to the representation of a specific target group within a specific intervention program. Building a profile is based on getting to know real people and helps us better understand those who will benefit from the program. It can be used to make key design and functionality decisions during the design process or to make recommendations for future programs. Beneficiary profiling is also useful when a clear prediction of how beneficiaries will contact and engage with the program is needed. The profile includes the motivations, barriers, and, in a particular way, the “essence” of what a person is [29].

The profile of the optimal beneficiary is one of the most understandable deliverables that can be used to improve the beneficiaries’ experience and enhance the design of targeted interventions. Because aging in our current society is gradually increasing, it is of great significance to study services designed for the elderly and to explore the theoretical system of the design of elderly services. We approached the profile of an optimal beneficiary from an intersectional point of view. Intersectionality explores the ways that multiple disadvantaged statuses interact with each other to create challenges and to limit the power of the individual. Older people manifest a high degree of intersectionality and have fewer opportunities to learn due to associated comorbidities, usually low income, loneliness (widowed, single, or divorced), and isolation. A study conducted on a community in southern Georgia used intersectionality to better understand the challenges of having the combined statuses of being an older adult, living in an isolated area, and having limited financial resources. The results demonstrate that the multiplicative and intersecting statuses of the study population create challenges in many areas. The challenges of these intersectional statuses limit access to services in ways that each individual status did not, thereby compounding challenges [30]. The evidence of such a complex profile makes sure that everyone keeps focusing on the beneficiary and helps the stakeholders learn and understand intersectionality.

Because it is important to explain the factors influencing special service designs for the elderly [31], we created a profile of an optimal beneficiary through qualitative–quantitative validation, a mixed method focused on three main attributes of the beneficiaries: user goals, behaviors, and attitudes.

The purpose in creating a profile of the optimal beneficiary became necessary as a result of the task of creating a yoga course adapted to the needs of the beneficiaries of the project, older people. In order to design a suitable material, we noted the need to know the typology of the beneficiary who will benefit the most from this program as much as possible. In addition to this aspect, there is the innovation of the proposal that comes, at least for Bucharest, with a different way of intervention for an age group characterized by a higher potential of rigidity for the new. Regarding yoga, the Romanian population still holds stereotypical beliefs, so understanding both the negative and positive beliefs older people have related to such a course was the first aim of the project. It was also particularly useful in helping us create realistic beneficiary’s journeys, who told the story of how they would come in contact with and engage with our program.

The general objective of this research was to create a specific profile of the characteristics of an optimal beneficiary of a newly developed program targeting an increase in the social inclusion and the active participation in social life of older adults. We wanted to communicate what the potential user experience could be to stakeholders, colleagues, delivery partners, and anyone else involved.

## 2. Materials and Methods

### 2.1. Study Design and Participants

This study is based on a mixed-methods research design, incorporating both quantitative and qualitative approaches. The quantitative part followed a cross-sectional design, while the qualitative component used a phenomenological approach through focus groups.

The study included two groups.

I.Quantitative group: A total of 50 hospitalized older adults who completed standardized assessment tools. Sampling was non-probabilistic, with participants recruited from the “Ana Aslan” National Institute of Gerontology and Geriatrics (NIGG).II.Qualitative group: Four focus-groups consisting of 17 participants (10 subjects from NIGG “Ana Aslan” and 7 subjects from GNSPY).

Demographic and health characteristics were collected separately for each group and they followed the below model:For the quantitative group: age, level of education, cognitive function (assessed by CASE-SF), mobility status (categorized as independent, requiring a cane, or requiring a walker), and use of assistive devices (hearing aids, glasses, mobility aids).For the qualitative group: age distribution, educational attainment, history of yoga practice, cognitive function, mobility status, and perceived physical limitations.

The selection criteria of the study participants were as follows.

#### 2.1.1. Inclusion Criteria

A.Demographic variable:
Adults over the age of 65 years;Living in underserved areas of Bucharest (areas in which public transportation is not easy to access);Low incomes (pensions under RON 3000/month).B.Social variable:
Loneliness (widower, single, divorced);Physical deficiencies, such as arthrosis, rheumatism, and reduced mobility;Mental dysfunctions, such as anxiety, depression, fear of aging, and impaired cognitive abilities.

#### 2.1.2. Exclusion Criteria

Severe cognitive impairment (e.g., diagnosed dementia);Severe mobility limitations (e.g., bedridden individuals or those requiring full assistance for movement);Uncontrolled chronic conditions (e.g., hypertension, advanced heart failure);Recent major surgeries (within the past three months);Severe psychiatric disorders (e.g., schizophrenia, severe bipolar disorder);Lack of informed consent due to cognitive or legal incapacity.

The selection process respected the inclusion criteria mentioned in the project: people over 65 years of age with a low financial income from underserved areas of Bucharest who also associate a somatic and/or mental pathology. Information regarding medical conditions was completed from the patient’s Observation Sheet, respecting the right to confidentiality and having their consent. The selection of participants, according to the demographic variable regarding the less served areas of Bucharest, was carried out, respecting the map of public transport network in Bucharest.

### 2.2. Ethical Considerations

The study was conducted in accordance with the ethical principles outlined in the Declaration of Helsinki. The ethical approval was obtained from the National Institute of Gerontology and Geriatrics “Ana Aslan” Ethics Committee (Approval Code: 178/23.11.2023). All participants were informed about the purpose and methodology of the study and provided written informed consent before their participation. Confidentiality and anonymity were maintained throughout the study. Personal data were anonymized and stored securely, accessible only to authorized researchers. The study complied with General Data Protection Regulation (GDPR) standards for data protection.

### 2.3. Measures

This study employed a mixed-methods approach, and the assessment tools were selected accordingly to evaluate both quantitative and qualitative aspects.

WHOQoL-BREF (World Health Organization Quality-of-Life Scale—Short Form):It assesses physical, psychological, social, and environmental health-related quality of life [32,33,34]. A higher score represents a higher quality of life [35,36].CASE-SF (Clinical Assessmenr Scales for the Elderly—Short Form):It assesses anxiety, depression, somatization, fear of aging, and cognitive competence. Each subscale score is derived by summing the individual item scores and normalizing them to a standardized scale, where higher scores indicate greater impairment [37].Subjective reports:Medical diagnoses were extracted from clinical records, not self-reported by participants. These diagnoses were not used as a numerical score but rather as contextual health information for participant profiling.Focus Groups:Explored perceptions, barriers, and motivational factors regarding yoga participation were measured. The 17 individuals in the focus groups were not part of the quantitative sample but were selected separately to assess their interest in yoga before any exposure to the intervention.

The participants’ willingness to take part in a yoga course was measured with one 5-point Likert scale item asking whether they would like to participate in such a course or not. Higher scores indicated a higher willingness to join the course.

### 2.4. Procedure

#### 2.4.1. Quantitative Data Collection

The quantitative assessments were conducted in a controlled clinical setting at the “Ana Aslan” National Institute of Gerontology and Geriatrics (NIGG) by trained clinical psychologists. The participants were individually evaluated in a quiet environment to ensure focus and minimize distractions. The standardized assessment tools, including WHOQOL-BREF and CASE-SF, were administered by trained clinical psychologists with expertise in geriatric assessment. Each session lasted for approximately 45–60 min, and the participants were encouraged to ask questions to clarify any doubts regarding the assessment process. The data collection followed a structured protocol to ensure consistency across participants.

For the quantitative analysis, after signing the consent form and filling in the recruitment form by each participant, the battery of tests was administered.

#### 2.4.2. Qualitative Data Collection

The focus groups were conducted in a dedicated meeting room at NIGG, ensuring privacy and a comfortable discussion setting. The sessions were moderated by experienced gerontopsychologists with expertise in qualitative research.

In order to respect the dynamics of the groups of beneficiaries, the selection included 10 people who had no contact with yoga and 7 people who currently practiced or had previously practiced yoga. Two focus groups were organized at “Ana Aslan” NIGG, whereas the other two focus groups at the GNSPY headquarters, who are our partner in this Erasmus project. For participation, the same selection criteria were followed, and an Informed Consent agreement and a Consent agreement for the audio recording of the focus group sessions were signed.

The first and third focus groups took place at the “Ana Aslan” headquarters and involved the voluntary participation of six and, respectively, four participants admitted to the clinic. The recruitment respected the inclusion criteria provided in the Erasmus application, and the participants were not familiar with the concept and had not participated in any yoga session previously. Discussions with each focus group were about 50 min.

The second and fourth focus groups took place at the GNSPY headquarters and involved the voluntary participation of three and, respectively, four participants, among the partner’s trainees. The recruitment respected the inclusion criteria provided in the Erasmus application, but the participants were either familiar with the concept of yoga or were senior practitioners. The durations of the focus groups were about 1 h and 30 min, respectively.

The selection was based on the extent to which the participants were (in the case of participants practicing yoga) or were not (in the case of participants from “Ana Aslan” NIGG) familiar with the concept of yoga and what it entails. For this reason, the following points were added the inclusion criteria: people over the age of 65 with reduced opportunities in terms of financial income, health damage, loneliness, and isolation.

Each focus group session followed a semi-structured guide (Table 1) covering topics such as perceived benefits, barriers, and motivational factors for yoga participation. The sessions were audio-recorded and transcribed verbatim and moderated by experienced gerontopsychologists. Field notes were taken to capture non-verbal cues and group dynamics

#### 2.4.3. Ethical and Confidentiality Considerations

All data collection adhered to strict ethical guidelines. Participants provided written informed consent before participation, and confidentiality was ensured by anonymizing transcripts and removing identifiable data. Only authorized researchers had access to the raw data, which was securely stored in compliance with General Data Protection Regulation (GDPR) standards.

### 2.5. Analysis

#### 2.5.1. Quantitative Analysis

The main objective of the quantitative analysis was to investigate the potential predictors of subjects’ willingness to participate in yoga classes. The presence of physical or mental pathology prior to hospital admission were treated as binary predictors.

Age, levels of anxiety, depression, somatization, fear of aging, cognition, and quality of life, as assessed by the questionnaires, were the continuous variables of the quantitative analyses. For ease of interpretation, two regression models were run, one for the categorical variables and one for the continuous variables. All statistical analyses were conducted using SPSS version 20.

#### 2.5.2. Qualitative Analysis

The objective of organizing these focus groups was to analyze the motivation to participate in a yoga program and the extent to which it aligns with the goals of the project. Existing stereotypes related to yoga were also identified and probed.

Conducting qualitative analysis followed the pattern of identifying themes and related motifs, a pattern that has become popular over the past ten years in research. It illustrates patterns by which ideas are subordinated to a general concept and by which analytical observations are thus organized. The theme expresses something meaningful about the information obtained and tells a story. Starting with these characteristics, the structure of the focus groups followed a certain logic to identify some central themes and the reasons included. The meetings were audio recorded which were then manually transcribed. The thematic analysis was started by two experienced researchers who analyzed the text and found the main codes. The raters then collapsed the codes, and, through open discussions, they agreed upon the main emerging themes. Theoretical saturation was achieved. The analysis was conducted based on the study of Braun and Clarke (2021) [38].

## 3. Results

### 3.1. Quantitative Results

In total, 50 subjects hospitalized at “Ana Aslan” National Institute of Gerontology and Geriatrics (NIGG)—Central Headquarters (Bucharest) completed the Questionnaire for the Assessment of the Quality of Life (WHOQOL-BREF), the Clinical Assessment Scales for the Elderly (CASE-SF), and answered the question related to the availability to participate in a specific yoga program.

Data set: N = 50; from the point of view of income level, 10% had a pension less than EUR 300, 46% had a pension between EUR 301 and 400, and 44% had a pension over EUR 400. In terms of social status, 74% were widows/widowers, 6% were single/separated, and 20% were divorced.

The assumptions for the regression analyses seem to have been met. Correlations between predictors were between +0.9 and −0.9, and VIF values were less than 10 (ranging from 1.04 to 6.18). Tolerance values were over 0.2, except for the fear of aging (tolerance 0.16) and the cognition (tolerance 0.17) variables.

The regression model between the continuous predictors and the outcome variable was not statistically significant (*F*(7, 42) = 1.33, *p* = 0.261), with the predictors explaining 0.05% of the variance in the subjects’ willingness to participate in yoga classes. β coefficient, t value, and level of statistical significance obtained from the regression model, as well as means and standard deviations for each predictor, are presented in Table 2. Age, levels of anxiety, depression, somatization, fear of aging, cognition, and quality of life did not significantly predict willingness to participate in yoga classes.

The regression model, including the categorical variables of having physical or mental diagnoses prior to hospitalization, was statistically significant (*F*(2, 47) = 5.44, *p* = 0.007), with the predictors explaining 15.4% of the variance in the subjects’ willingness to participate in yoga classes. The β coefficient, t value, and level of statistical significance obtained from the regression model, as well as means and standard deviations for each predictor, are presented in Table 3. A priori physical diagnoses are correlated with an increase in subjects’ willingness to participate in yoga classes, while a priori mental diagnoses are correlated with a decrease in willingness to participate.

### 3.2. Qualitative Results

A thematic analysis was conducted based on the transcripts of the four focus groups, and five dominant themes were identified, which, in turn, include a series of related codes. They provide a valuable window into how subjects perceive and relate to yoga (Table 4). 

(1)The Health theme includes a series of codes on three significant levels, namely physical, cognitive, and social.On a physical level, the subjects associate the idea of yoga with benefits in the sense of acquiring extra energy, protection against diseases, and performing slow movement adapted to the needs of older people. Codes with a negative role refer to the existence of a severe pathology with a disabling role. At the cognitive level, the idea of obtaining an improvement in mnesic and prosexic functions was identified. At the social level, the reasons are related to the opportunity for interaction and the inclusion of older people in groups.(2)The Information theme contains a series of reasons regarding the roles that information had when potential beneficiaries chose to participate. These roles both reinforce and block willingness to participate.As a positive impact, the thematic codes center around the idea that yoga includes varied and different forms, some of which are adapted to the dynamics of an older person and the support of participation from family and/or friends in this endeavor. As a blocking role, the information held is related to the sexualization of the concept of yoga and, as a consequence, to the presence of stereotypes and prejudices.(3)The Flexibility theme includes the reasons related to certain personality traits with the role of increasing the willingness to participate or cancel this openness.Thematic motifs that positively impact the willingness to participate in yoga classes refer to traits such as openness to the new, curiosity, not caring what others might say, and putting oneself first. Traits with a negative impact include resistance to change, convenience, and reluctance to new experiences [39].(4)The Organization and Distribution of Resources theme includes thematic reasons with a rather negative role in openness to participation.Within this theme, responsibility and roles in the extended family, a certain dynamic of personal life, limited financial resources, and reduced physical and spatical accessibility were identified as the general reasons that impact the willingness to participate in yoga classes. How a person is defined and how they relate to their roles lead to a certain identity and can have both positive and negative impacts.(5)The theme of Identity refers to those leitmotifs that describe certain attitudes and behaviors arising from personal history.

Also here, the adaptive mechanisms that were developed are included, focusing on those that have a positive or interfering role with the willingness to participate in yoga lessons.

These are the desire for knowledge, development, and belonging, the presence of resilience resources to contain trauma, and the need for a paradigm shift constitute those identity traits that associate a positive imprint. However, strong religious beliefs may conflict with the tendency to participate or even limit access.

The visual translation of these themes provides the image of reciprocal connections that support and reinforce their role in relation to their willingness to participate in yoga lessons (see Figure 1).

The information obtained from the qualitative and quantitative analysis helped shape a portrait of older people who would benefit most from a specific yoga course. The profile answers questions related to the specifics of these people from the points of view of motivations, limitations, and personal traits, and the way in which they correlate with the objectives of the project and with the particularities of yoga practice.

The extent to which the goals of the older person are associated with the goals of the project:

A specific problem that older people face is related to loneliness and the presence of feelings of isolation and loneliness. EU-LS 2022, the first EU-wide survey on loneliness, shas found that, on average, 13% of respondents report that they have felt lonely most or all of the time during the past four weeks, while 35% have reported feeling lonely at least part of that time. The incidence of these feelings differs from one country to another, being higher for countries in Southern and Eastern Europe compared to those in Northern and Western Europe. Respondents from Romania and Italy are both at a moderate level [40].

In Romania, according to a national study carried out by Kantar Romania in 2021 regarding the loneliness index, 1 in 4 elderly people in the urban environment, that is, over 450,000 people face a high degree of loneliness, and 36% feel a medium degree of loneliness. Furthermore, 32% feel marginalized and live in continuous isolation [40].

Feelings of loneliness are associated with impaired physical and mental capacities. The results show that individuals who feel lonely most or all the time are three times more likely to rate their health as poor. Single people are also more likely to be depressed and tend to engage more in unhealthy behaviors, such as habits that lead to addiction [41].

One of the most important themes that emerged during the focus group is related to health, social health, and the need for constant social interactions.

This condition of loneliness is important when it is not associated with the presence of mental disorders such as anxiety, depression, and cognitive impairment. The results of the quantitative analysis indicate that the present psychological dysfunctions act as a barrier in terms of participation in yoga lessons.

The way in which an older person’s motivations are associated with the specific elements of the project, namely the development of the competence of awareness of one’s own body, breathing, emotions, and thoughts, is adapted from yoga techniques in order to increase the quality of life.

The presence and education of this competence aims at the ability to introspect, effectively manage time and information, collaborate with others in a constructive manner, and remain resilient. Thus, an increase in the quality of life is achieved by developing the ability to lead a self-aware and future-oriented life to contain complexity and uncertainty, and to learn about and maintain physical and mental health. Quality of life assessment refers to how a person evaluates their life. It represents a broad, reflective assessment that a person makes of their life.

According to the Eurostat’s 2022 survey on overall life satisfaction by age group, in most EU member states, the 16–29 age group indicated a higher life satisfaction compared to the 65+ group. The difference between the two age groups is higher in the case of Romania compared to Italy, but not by much (—1.0 points versus 0.6 points) [42].

The qualitative analysis emphasizes the importance of older people participating in the focus groups give to increasing their quality of life. This leitmotif is found in the four central themes and in the perception of both categories of participants: yoga practitioners and people who have never participated in yoga before.

Identifying the general attitude about yoga by exploring existing stereotypes:

While globalization makes it possible for different parts of the world to share traditions and cultures, yoga has not only been shared but has also undergone several changes to be assimilated into the Western world. Yoga was not and is still not popular in countries with a deeply Christian history. Academic studies that have explored how contemporary practices often deviate from traditional intentions influenced by cultural and media dynamics recall a sexualization of yoga in Europe and the United States. Limited or exclusionary media representations may discourage individuals who do not fit the representative models of yoga in terms of gender, age, race, or body type [43]. There is also the idea that yoga classes are expensive as there might be a need for specific equipment among other things.

Exploring existing stereotypes in the case of people over the age of 65 is important in identifying the barriers that could intervene in the manifestation of a willingness to participate in yoga. The qualitative analysis indicates the possibility that a series of specific stereotypes and prejudices associated with the theme “Information”. The reasons that constitute barriers and that belong to this theme are related to the belief that yoga involves a series of complex physical movements with a high degree of difficulty for older people. Aspects related to limited financial resources were also mentioned, as the subjects associated yoga with high costs.

The existence of some general prejudices regarding yoga in Romania that have their origins in a highly publicized criminal case from the 2000s have also impacted the way in which are subjects perceived yoga [44].

Identifying the degree of mental flexibility, openness, and availability to new things:

Mental flexibility adds important elements to understanding how a healthy, self-managing person function in an uncertain, unpredictable world around them, where novelty and change are the norm rather than the exception. Psychological flexibility spans across a wide range of human abilities from adapting to various situational demands, including checking stereotypes or behavioral registers when these strategies compromise personal or social functioning; maintaining balance between important areas of life; being aware; and showing availability [39].

The motifs for “Flexibility” and “Identity” indicate the importance of some personality traits and some psychological constructs that may increase people’s availability to get involved in yoga classes [45]. The presence of these features is important, because in relation to the novelty and stereotyping of the concept of yoga for the elderly population, some of them can act as obstacles. A series of reasons, such as “convenience”, “reluctance to new” in the case of the theme “Flexibility”, and elements of identity spirituality in the theme “Identity”, are included here.

Depending on the existence of some personality traits with the role of manifesting and strengthening availability, the optimal beneficiary is a person who presents a moderate–high degree of mental flexibility. The person is present: “If we dwell on the past or focus on the future, we focus on things that are out of our control, but also if we react, then we may not act or make decisions based on our values, beliefs, and goals”. The person shows availability: Flexibility requires people to be open to new experiences and perspectives. The person shows acceptance: Flexible people are able to accept what they feel, acknowledge their emotions, and look for ways to create meaning and grow.

### 3.3. Profile of the Optimal Beneficiary

The captured features for profile of the optimal beneficiary are as follows:(1)Socio-demographic characteristics
A single person: divorced, widower, single;A person with physical dysfunctions: arthrosis, rheumatism, impaired mobility;A person who does not have mental disorders such as anxiety, depression, moderate-major cognitive impairment;A person from the urban environment from insufficiently served peripheral areas;Older people: over 65 years.(2)Psychological characteristics
A person who has an impaired quality of life and/or wishes to increase it;A person who has formed a perception and belief about yoga based on correct, undistorted information;A person who has received strengthening information about yoga over time from adjacent sources, such as family, friends, teachers;A person who checked their existing stereotypes through an informed check and who did not overgeneralize a negative event;A person who presents moderate mental flexibility;A person who shows openness to new things;A person who focuses on the present;A person with a spiritual identity that does oppose to yoga practices.


## 4. Discussion

Identification of a profile for the optimal beneficiary was necessary because of the task of creating a Syllabus of 12 yoga lessons adapted to the needs of the target group—older people with reduced possibilities. In order to prepare an adapted material, we needed to know the typology of the beneficiary who would benefit the most from this program as much as possible.

The reasons related to the five themes identified following the qualitative analysis outline a series of barriers that limit participation in yoga. The limitations discovered following the qualitative analysis converge with those described in the specialized literature.

Health motivation is associated with the behavior of being informed and concerned about health status at all levels. The reasons related to health become an obstacle only in the presence of severe somatic conditions. The information people have about yoga acts as both an enabler and a constraint. Given that this theme is related to health and flexibility, the role of information is one of strengthening people’s willingness and availability to get involved in yoga classes. The presence of distorted and incomplete information associated with reduced flexibility has a limiting role. Personality traits connected with mental flexibility lead to an increase in interest in yoga. Rigidity towards present roles and the existing organization of life, together with convenience and reluctance to new experiences limit people’s willingness to participate in yoga classes.

Depending on the way we defined the themes and especially the theme of identity, the qualitative analysis reveals that a certain identity structure characterized by mental flexibility, openness to correct information, and health motivation on all important levels leads to an increase in the willingness to attend sessions of yoga.

According to a 2020 study of 37 pre-frail and frail older people at a medical center examining their beliefs about yoga and identifying barriers to participation, the main barriers to practicing yoga were perceived difficulty of yoga practice, lack of motivation, and fear of injury [46].

A dissertation published in 2013 aimed to explore the modalities, motivations, barriers, and experiences of 452 yoga session participants in two age groups (40–55 years and 55+ years) and to detect potential differences identified a number of interesting aspects. Both age groups ended up practicing yoga in similar ways; through outreach, reading yoga books, and having a friend or family member suggesting yoga. Exploring the motivations for practicing yoga revealed differences between the two age groups. Middle-aged subjects are motivated by stress and anxiety reduction, weight loss, and increased muscle strength. Older people were motivated by physical health issues to prevent osteoporosis, improve flexibility and social health, and increase social interactions. The barriers identified in the case of subjects over the age of 55 were related to costs and accessibility [47].

A 2011 study that sought to assess the self-reported perceptions and effects of yoga in 12 older adults identified the perceived benefits as improved gait and balance, decreased pain, decreased need for medication, reduced stress, improved sleep, relief of anxiety and depression, increased mobility, increased self-awareness, and a greater sense of peace. Limitations to participation that emerged included the subjects’ daily obligations, accessibility, and the interference of religious beliefs. These hindered the subjects’ ability to fully participate [48].

### Strenghts and Limitations

Our project’s main objective is to further the scientific knowledge on the effects of yoga on the competence of self-awareness and to achieve significant improvements of the practice with older adults. Moreover, the effectiveness of the program can lead to more inclusiveness of older adults within the local communities, which, in their turn, can get integrated in the wider European communities.

The study presents several important limitations.

First, the whole sample was selected non-probabilistically, meaning that participants were not randomly chosen from a larger population but rather based on predefined criteria. This may affect the generalizability of the results, as the studied groups may not be representative of the entire elderly population.

Second, the number of participants in the focus groups was relatively small (17 individuals), which limits the diversity of perspectives captured. A small participant group may introduce bias in the interpretation of qualitative data, as not all categories of older adults interested in yoga may be adequately represented. Additionally, self-reported data on quality of life and mental health may be influenced by subjective factors, such as the participants’ emotional state at the time of meeting.

## 5. Conclusions

Considering the identified issues of loneliness and its consequences, the general objective of our project has been to increase inclusion and participation in social life as well as to implement measures to combat it. The optimal beneficiary of the project is a lonely and isolated person with limited social interactions but without mental disorders such as anxiety, depression, or cognitive impairment.

Additionally, considering the motivations of older adults and the importance of improving their quality of life, the optimal beneficiary is someone experiencing a decline in their quality of life and seeking to enhance it. A deterioration in quality of life at the somatic level correlates with a higher willingness to participate in yoga lessons, whereas psychological impairment tends to act as a barrier.

Depending on the presence of stereotypes and prejudices at the collective and personal mental level, the optimal beneficiary is considered to be the person who formed a perception and belief about yoga based on correct, undistorted information or who received reinforcing information about yoga from, over time, other sources, such as family, friends, or books. Also, another optimal beneficiary is a person who has verified their present stereotypes through an informed verification and who has not overgeneralized a negative event.

Lastly, personality traits play a crucial role in fostering and strengthening an individual’s openness to change. The optimal beneficiary is a person with moderate to high degree of mental flexibility, capable of staying present and making decisions aligned with their values and goals. Such a person demonstrates openness to new experiences and perspectives, accepts their emotions and seeks ways to create meaning and personal growth.

The integration of adapted yoga programs into geriatric care can have a positive impact on the physical and mental health of older adults, contributing to improving mobility, balance, and overall well-being. From a clinical perspective, we recommend these programs to be included in the care strategies for older adults. Healthcare professionals could collaborate with yoga instructors specialized in therapeutic approaches to personalize exercises to each patient’s needs.

## Figures and Tables

**Figure 1 geriatrics-10-00059-f001:**
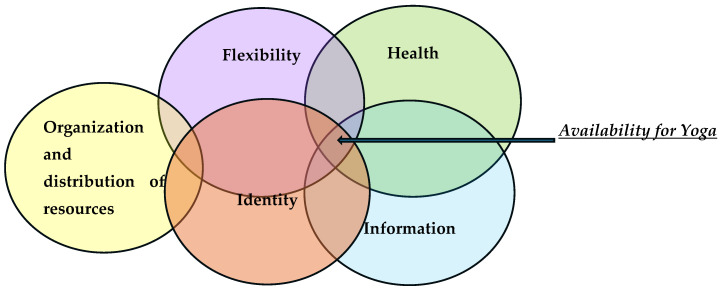
Graphical visualization of dominant themes.

**Table 1 geriatrics-10-00059-t001:** Focus Group Question Guide.

I. Introduction and Welcome
Ethics and confidentiality;
Purpose and theme presentation;
The story of the name;
Group rules.
II. Introduction Questions
Objective: The debate initiation and participants familiarization with the dynamics of the interaction.
- *When did you first hear about yoga?*
- *Where did you find out and/or from whom?*
- *What is the message that you received?*
III. Key questions
Objective 1: Identification of participants opinion about yoga.
- *What do you know about yoga now?*
- *How familiar is the concept of yoga in your personal, family, social environment?*
- *What do you think yoga is for?*
Objective 2: Determination of the usefulness level of a yoga program for increasing and promoting well-being and inclusion stimulation.
- *Do you think that a sustained program of yoga lessons could bring a benefit in your life now?*
- *What benefits are you thinking of?*
Objective 3: Identification of reasons why participants would or would not follow a yoga program.
- *What are the reasons why you chose to participate or not to participate in this program further?*
- *What do you think there are the reasons why other people would not participate?*

**Table 2 geriatrics-10-00059-t002:** Regression model between the predictors of physical and mental diagnoses and willingness to participate in yoga classes as the outcome variable.

	*M*	*SD*	β	*t*	*p*
Age	73.06	5.63	−0.07	−1.66	0.105
Anxiety CASE-SF	38.42	10.75	−0.04	−1.42	0.164
Depression CASE-SF	40.86	9.88	−0.05	−1.13	0.267
Somatization CASE-SF	38.86	9.10	−0.02	−0.36	0.720
Fear of Aging CASE-SF	41.21	13.36	0.06	1.58	0.122
Cognition CASE-SF	44.90	13.28	−0.05	−1.21	0.232
WHOQOL-BREF	99.32	15.38	−0.04	−1.92	0.061

**Table 3 geriatrics-10-00059-t003:** Regression model between physical and mental diagnoses as predictors and willingness to participate in yoga classes as outcome.

	*M*	*SD*	β	*t*	*p*
Physical diagnosis	0.12	0.33	1.65	2.73	0.009
Mental diagnosis	0.78	0.42	−1.12	−2.36	0.023

**Table 4 geriatrics-10-00059-t004:** Examples for each theme that occurred during the focus groups.

Theme	Codes	Quote
1. Health	Positive: Acquiring extra energy Protection against diseases Physical exercise Social inclusion of elderly people Opportunity for social interaction Negative: Severe debilitating pathologies	“Being several people we communicate, it’s different than sitting alone and thinking, you communicate, one says one thing, one says another...”
		“Socialization, yoga sessions open your horizons a bit and you socialize better... And the brain, you still have contact with one, with another. If you stay locked in a room being old, not having contact with so-and-so, you don’t talk to anyone anymore...”
		“I felt that I entered a community. So it was a matter of emotional compatibility, so I say that the spiritual part also worked a lot for me...”
		“An inclusion in a community that has maintained and developed beautifully and even supports me in times of trouble…”
		“Me because I have a tumultuous life, I have friends, I have children, I have children who take special care of me, my life is not static, not “oh, is someone calling me?”, it seems to me that it would be the same with socializing…”
		“It relaxes the mind, the body, and at our age I think it’s very good...”
		“And for mind, because the body has aged, the mind has also aged, you no longer think like you did when you were young, you used to think seven, now one and you remember after I don’t know how long and for memory more would be...”
		“For the benefit of the memory, I’m sure that the meditations there helped a lot the memory, any meditation helps a lot the memory...”
		“I think it’s good for the health, it balances the psyche and it’s also a pleasure after all...”
		“I noticed that I felt better, I wasn’t so dizzy anymore, I didn’t trust my legs anymore, they were shaking. Now these symptoms have started to disappear...”
		“I think that during these years, it was what balanced me and supported me because I needed something like that and so it was, not only physical, that I always liked to move, so I enjoyed what I did of...”
		“I was going to work mentally, to understand something from this life that basically we all have, a search and there was also another family...”
		“I learned a lot from yoga to be observant, to observe yourself, so that was essential in life. There were some life criteria that guided me later...”
		“It gave me a balance here and this attention to myself and to those around me helped me a lot...”
		“For me it meant a great awareness, the very first time I realized that I need to be aware of some things both physically and psychologically. Social really meant that too, an inclusion in a community that has maintained and developed beautifully and even supports the times when I’m in trouble...”
		“I would get up in the morning, do my yoga routine, and suddenly I was a different person. I had energy, I had how to say, power and mental.”
2. Information	Positive: Varied and different forms of yoga Support from family and friends Negative: Sexualisation of yoga Stereotypes Prejudices	“About yoga, I was left with a bitter taste, why, I received the information that this Mr. Bivolaru was recruiting young people, raping them, forcing them to be there, this was the message I received...”
		“I remembered that I was talking with a colleague about yoga and the first thing she said was “oh, I’m not going, because it seems like I’m having group sex!”. Speaking of Bivolaru. So the world was left with a negative perception...”
		“If they hear about Bivolaru and you say you do yoga, the world can say—who knows what he did over there, that’s why he went—Romanian mentality!...”
		“I heard about yoga when there was that conversation with... Guru, what does he call it...”
		“In the area where I live, there was Bivolaru, in a tall house with one floor, there were always blinds shot, all the time, people said at the beginning it was prostitution, people said they were making some sexy videos, but no one knew what was going on there...”
		“I first heard from my children about it, they had a teacher who went at this classes, he liked it, went...”
3. Flexibility	Positive: Openness to new experiences Curiosity Prioritizing oneself Not caring what others say Negative: Resistence to change Convenience Rigidity to new experiences	“It’s a matter of laziness and fear to start something...”
		“For me, for example, it’s a necessity. For us, the necessity is in the family, we live in a group, with the grandparents, so basically you allocate your time in what you are actually attracted and of course you feel good too, I mean each of us has this job of doing something for others... I didn’t have this and I turned to yoga...”
		“There are some habits, I made my schedule, I drink my coffee, I have to go to the toilet and only after that do I start my schedule at a certain time. So it’s those habits we get into that we don’t want to get out of...”
		“I didn’t agree with yoga because I personally have a slightly different body than the majority. I have some stuff that doesn’t fit the standard. And I focused on religious thinking...”
		“I am going to parallel and to the church and coming and in yoga, I think I found the meaning more on the other side, and here I found support and balance...”
4. Organization and distribution of resources	Positive: Roles within the extended family Negative: Limited financial resources Reduced accessibility	“I don’t know if my health would allow me. I have problems: with my heart, with my gland, with my liver, if I make ten steps faster, my blood pressure goes up. And then I don’t know if I could participate in any classes...”
		“I might not be able to do certain things and that’s the only reason that would stop me...”
		“I’ve talked to some people who consider themselves too old, so 72-year-olds who already think they’re no longer capable, to do something...”
		“There are also financial aspects that would impact...”
		“The lack of correct information. That is, when they see that there are only ads with sexy girls, without... and they don’t see people moving either a little more...”
5. Identity	Positive: Willingness to evolve Need for belonging Need for a paradigm shift Resilience resources Negative: Rigid religious beliefs	“Since I also go to church, I think I find more meaning there, while here I found my support and balance...”
		“For me it started from a personal need. Traditionally, this need is fulfilled within the family, where we live as a group—with grandparents, nephews, and so you are basically necessary. Therefore you allocate your time to something that attracts you and makes you feel good, so each of us has this need to do something for others... I did not have that, therefore I started doing yoga...”
		“I did not agree with participating in yoga activities, because I have a bit of a different body when compared to the great majority of people. I have some traits that do not fit the standard. As a result, I centred myself on religious thinking...”

## Data Availability

The dataset used and analyzed during the current study is available from the corresponding author upon reasonable request.

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
