# Peer review of "An Optimal Beneficiary Profile to Ensure Focused Interventions for Older Adults"

_geriatrics, 2025, doi:10.3390/geriatrics10020059_

Round 1

Reviewer 1 Report

Comments and Suggestions for Authors

First of all, I would like to congratulate the research team. I hope that my comments will be a contribution, allowing them to present an even more solid article for publication.

I detail some aspects to improve in general, and in each section of the scientific article below:

General comment:

  • I found the article difficult to understand. I felt like I had to go back and forth to different parts of the text to understand it, particularly in the Materials and Methods, Results, and Discussion.
  • Further development of the introduction and precision of the methodology is lacking.
  • In addition, there are results that are located as discussion, and vice versa.
  • I think the article would benefit from making it explicit that it is a mixed methods study, and addressing the methodology and results by presenting the quantitative part first and then the qualitative part. In addition, the triangulation of both approaches (and their results) should appear at the end of the results and in the discussion.
  • Recently, the journal published a mixed methods article. This may guide authors in addressing some of the recommendations I raise below. This article can be found at the following link: https://www.mdpi.com/2308-3417/10/1/8

Title:

  • It should be specified that it is a “Yoga Intervention”.
  • Also, it is a mixed methods research.

Abstract:

  • The abstract must explain the following sections: Background / Objectives , Methods, Results, and Conclusions.
  • I recommend adding 2-3 introductory lines that reflect background and/or the need to conduct this research. This should be done before the objective.
  • It should be specified that it is a mixed design, in addition to the specific design of the quantitative and qualitative parts.
  • It should be clarified where the 50 older people to whom the assessment tools were applied came from. Likewise, where the 17 older people to whom the focus group was applied came from (differentiating both subgroups).

Keywords:

  • I advise adding: Mixed methods.
  • I also recommend adding: Psychogerontology, and Mental Health.

1. Introduction: 

  • I suggest removing the 2 subtitles (Context, and Purpose and Objectives ), which will allow for a more fluid reading.
  • However, my comments are based on how the introduction is currently presented in the article.

1.1. Context 

  • I suggest adding a first stanza that contains:
  1. a) demographic background, in relation to the proportion of older people: worldwide, European and/or from the country of origin of the research.
  2. b) epidemiological background, which supports the practice of Yoga in older people in general: Risk Factors, pathologies or Geriatric Syndromes.
  • It is necessary to define what Yoga is, and the types of Yoga that exist. At the same time, which ones are most frequently used by older people, and reported in the literature.
  • It should be indicated in which situations Yoga is used/indicated (as an intervention) in older people, its effects (reported by other research) on biological, functional, psychological, social and spiritual health. Then, special emphasis should be placed on the inclusion and social participation of older people, which will later be the emphasis of the profile to be found in this research.
  • Similarly, indicate whether Yoga is contraindicated in any health condition in older people.
  • All of the above, before delving into the more specific aspects of psychogerontology in relation to Yoga in older people.
  • Also, the characteristics of Yoga (as an intervention) should be studied in depth in order for it to have effects on the mental health of older people, in terms of the minimum requirements (parameters): intensity, duration of a session, weekly frequency of sessions, duration of the intervention in months. To do so, I recommend searching for/referencing a systematic review and/or meta-analysis.
  • These last aspects (parameters) can also be added when describing the Erasmus+ Project, allowing for a better understanding of the ongoing project and the Yoga intervention with older people.
  • Also, it should be explained what adaptations are made to Yoga with older people and that were addressed in their project.
  • When describing the Erasmus+ Project, the objectives are mentioned. This creates some confusion with the objective of this research. Please review this aspect so as not to confuse the reader.
  • Line 119-120. “Numerous studies” are indicated, but there are no bibliographical references to support this.

1.2. Purpose and Objectives

  • The aim of the research is the last stanza. The rest is introduction.

2. Methods:

  • The title for this section should be: Materials and Methods
  • It should be indicated type and design of the research. Therefore, I recommend specifying that it is a “mixed methods” research, and specifically indicating the mixed research design used (with its respective bibliographical reference).
  • Then, in a stanza, specify the research design for the quantitative part, and then for the qualitative part. With its respective bibliographic reference for each design approach.
  • The two previous points will give a better understanding and replicability of the research.
  • I recommend using and reporting, as a supplementary table, some checklist for qualitative research (COREQ-32 or similar) of this research.
  • The ethical aspects of all this research may be better placed in this section. Evaluate.

2.1. Participants

  • I advise keeping the mixed design structure.
  • Therefore, it should be specified that the sampling was not probabilistic.
  • Also, it must be specified whether the older people in the group in which the focus groups were conducted (qualitative part) are a subgroup or not, of the group of 50 older people in the hospital in whom the evaluation scales were applied (quantitative part).
  • The sociodemographic characteristics that would be collected from the group of 50 people and from the group of 17 older people should be indicated separately. I also understand that in the 17 people there are 2 additional subgroups that were compared. This should be clarified.
  • In each group (50 and 17), the cognitive, functional and educational level of the participants must be specified. Also, indicate whether they used any type of technical aid (cane, walker).
  • The Ethics Committee that evaluated the project and the approval code must be indicated. Also, if the older people read, understood and signed the informed consent. This for both groups. Check if this is better at the end of 2.
  • Delete line 152.
  • Evaluate whether rows 148 to 150 are a better result, specifically in the first part of the sample characterization that I discuss below (3.).

2.2. Inclusion Criteria

  • I think this section should not be a title. Instead, it should be stanzas from the previous section.
  • Specify whether “disadvantaged areas” refers to areas with lower socioeconomic or educational status and/or higher crime rates.
  • Were there exclusion criteria?. These must be incorporated.

2.3. Measures

  • I advise keeping the mixed design structure.
  • Therefore, I would specify which assessment and/or evaluation tools were used for each part of the mixed design.
  • For each subscale of the CASE-SF tool, you must specify how the score is obtained and/or how it is analyzed. Also, the minimum and maximum score in each subscale.
  • It is not clear whether the “Subjective reports ” were diagnoses that were in the clinical record or were spontaneously reported by the older people. Were these used as a score?. These aspects need to be clarified.
  • When I get to the “focus groups” part, it is not clear to me whether the 10 people were part of the group of 50 or not. This needs to be clarified. I only understand now from reading the article that the 7 people alone are the ones who did Yoga.

2.4. Procedure 

  • I advise keeping the mixed design structure.
  • Therefore, detail them separately in this section. It is important to specify the conditions under which the evaluation tools (quantitative part) were applied, as well as the focus groups.
  • Who applied the assessment instruments?, Was it part of the research team?, Was it another person?, What was their profession?, What level of training did they have in applying the tools?. This should be specified.
  • Who applied the focus groups?, Was it part of the research team, was it another person?, What was their profession?, What level of training in applying focus groups did they have?. This should be specified.
  • Also, it should be specified what the Yoga intervention was like that the group carried out (I understand that they were part of the focus group). This is a key element of this research. Aspects such as (parameters) should be described in detail: type of Yoga used, intensity, session time, weekly frequency of sessions, duration of the intervention in months, adaptations made with the older people. I recommend adding a figure that reflects the Yoga intervention. This allows us to identify what Yoga experience these older people had.

2.5. Design and Analysis

  • It is possible that the first part of this section (rows 237 to 263) corresponds to the previous section, so I recommend relocating it. Also, that the information contained in these rows be placed as a main or supplementary table. And it should be specified as a “focus group question guide…”. Indicating the common questions and for each group. If there were some differentiated questions for those who practice Yoga and others for those who do not, they should also be added/differentiated.
  • Consequently, this section should simply be called “ Analysis ”, maintaining the same mixed method logic, with its respective description for the quantitative part first, and then for the qualitative part.
  • Consequently, in the quantitative part it is necessary to describe whether the normality of the data was analyzed, the statistical tests that were used, the level of statistical significance, and the statistical program used.
  • For the qualitative part, the focus groups were audio-recorded and transcribed in full and verbatim?, Field notes were taken from the focus groups?, The data were coded?, Who analyzed the information?. This should be specified.
  • Similarly, it is necessary to reference the author on whom they based themselves (and the steps they followed according to that author), to describe “ the pattern of identification Themes and related topics motifs ”.
  • Theoretical saturation was reached in the sample of focus groups?. This should be specified.
  • Was any program used for qualitative analysis?. This should be specified.
  • Consequently, it should be made clearer how we moved from story to text, codes, themes and subthemes.

3. Results:

3.1. Quantitative Analysis

  • As I mentioned before, a more detailed table is missing with the sociodemographic and health characteristics of both groups (50 and 17). And the respective subgroups in the case of the focus groups. Please, to be incorporated.
  • It seems that rows 282 to 285 correspond to some section of “ Materials and Methods”. Analysis ?
  • It was not used in the model 2 ( Table 2) the variable: “Perceived impairment of one's quality of life.” ?.

3.2. Qualitative Analysis

  • I recommend that the title of each topic be in italics, as is the case with the “Health” topic.
  • I think that somewhere in topic 4, the idea of “personal barriers to doing Yoga” should be outlined.
  • Figure 1 may benefit from adding to the figure the “levels” of the Health theme, as well as the “ leitmotifs ” identified in the various themes.
  • Textual citations should be associated with the results and not with the discussion.
  • Rows 360 to 384 correspond to discussion and should be placed there.
  • In rows 385 to 389 it is not clear what profile of older people is being referred to. I recommend carrying out a sort of triangulation between the quantitative and qualitative results, and explaining in the text what profile of older people would benefit most from a yoga intervention of these characteristics. Apparently this is in rows 563 to 584.
  • It is also not clear and/or explicit how these variables relate to the inclusion and social participation of older people. This can be discussed, however.

4. Discussion and Conclusions: 

  • This section should only be titled “Discussion”. I will leave a comment for the other section later.
  • I think that I would not generate 2 subsections (Structure, and Profile of the Optimal Beneficiary). Therefore, in my comments I only considered it as if it were a single section of discussion.
  • Rows 414 to 423 correspond to results. Apparently, they are several quotes that reflect the stories of several older people. Therefore, each quote must be separated, and identified as such (and according to groups) in the results.
  • Same as rows 451 to 471.
  • Same as rows 495 to 502.
  • Same as rows 506 to 518.
  • Same as rows 542 to 552.
  • This will generate a better flow in the discussion.
  • I would avoid the bullet points in rows 556 to 562, and keep it as one text in one stanza.
  • Towards the end, implications/guidelines for clinical practice should be added.
  • Likewise, implications/guidelines for future research.
  • The strengths and weaknesses should be added as a final stanza.

Conclusions: 

  • It should be developed as a separate section and respond to the research objective.

References: 

  • I request that a greater number of bibliographical references be added to support the research and its discussion. I recommend approaching 40-50 bibliographies (at least) to give greater robustness to this research. From 2020 onwards.
  • I recommend adding some bibliographical references, specifically systematic reviews, on the use of yoga in older people and/or its benefits on the health of older people.

Kind regards,

Author Response

1. Summary

We thank you very much for your valuable remarks and comments that led us to improve the quality of our manuscript. Please find the detailed responses below and the corresponding revisions/ corrections highlighted/ in track changes in the re-submitted files. 

2. Questions for General Evaluation

Reviewer’s Evaluation

Response and Revisions

Does the introduction provide sufficient background and include all relevant references?

Yes/Can be improved/Must be improved/Not applicable

3. Point-by-point response 3 (R1.2., R3.2., R3.3.)

Are all the cited references relevant to the research?

Yes/Can be improved/Must be improved/Not applicable

3. Point-by-point response 3 (R3.2., R3.3., R3.4., R.7, R9.4.)

Is the research design appropriate?

Yes/Can be improved/Must be improved/Not applicable

3. Point-by-point responses 4 - 9

Are the methods adequately described?

Yes/Can be improved/Must be improved/Not applicable

3. Point-by-point response 4

Are the results clearly presented?

Yes/Can be improved/Must be improved/Not applicable

3. Point-by-point response 10

Are the conclusions supported by the results?

Yes/Can be improved/Must be improved/Not applicable

3. Point-by-point response 11 (R11.3.)

3. Point-by-point response to Comments and Suggestions for Authors

Comments 1: General comment: 

Response 1: We agree with these comments.

C1.1. I found the article difficult to understand. I felt like I had to go back and forth to different parts of the text to understand it, particularly in the Materials and Methods, Results, and Discussion.

R 1.1. Therefore, we have developed the introduction section, we clarified the objectives of this article, we rebuild the structure of methodology section, we added tables, we rearranged sections of results and conclusions and completed the bibliographic part.

C1.2. Further development of the introduction and precision of the methodology is lacking.

R1.2 We developed the introduction by adding demographic and epidemiological information, the definition of yoga and types of yoga, the effects of yoga on physical and mental health, and clarified the purpose of this article. We have rebuilt the structure of the Methodology.

C1.3. In addition, there are results that are located as discussion, and vice versa.

R1.3. We have corrected the existing inaccuracies.

C1.4. I think the article would benefit from making it explicit that it is a mixed methods study and addressing the methodology and results by presenting the quantitative part first and then the qualitative part. In addition, the triangulation of both approaches (and their results) should appear at the end of the results and in the discussion.

R1.4. We have specified that we used a mixed method study design as an analysis method. We have made the sequence of steps in the methodology and results sections clearer by respecting the order of quantitative analysis and qualitative analysis, respectively. The result of the two approaches - the optimal beneficiary profile - we moved it to the end of the results section and based on it we developed the discussion section.

C1.5. Recently, the journal published a mixed methods article. This may guide authors in addressing some of the recommendations I raise below. This article can be found at the following link: https://www.mdpi.com/2308-3417/10/1/8

R.1.5. Thank you this sugestion, it was of most help.

Comments 2:

Response 2: Thank you for your comments.

C2.1. Title: It should be specified that it is a “Yoga Intervention”.

       Also, it is mixed methods research.

R2.1. “Yoga Intervention” will be included in a future paper, namely the main paper of the Erasmus project as they do not fit the objective of this study. On page 1 we specified that it is “[mixed design]” in Abstract (line 12) and we added “[mixed methods]” at Keywords (line 26).

C2.2. Abstract: The abstract must explain the following sections: Background / Objectives,  

      Methods, Results, and Conclusions.

I recommend adding 2-3 introductory lines that reflect background and/or the need to conduct this research. This should be done before the objective.

It should be specified that it is a mixed design, in addition to the specific design of the quantitative and qualitative parts.

It should be clarified where the 50 older people to whom the assessment tools were applied came from. Likewise, where the 17 older people to whom the focus group was applied came from (differentiating both subgroups).

R2.2. On page 1 we have divided the abstract based on the 4 sections [Background, Methods, Results, Conclusions], and we added the introductory lines reflecting the background .[Background: Aging is a lifelong process, and many chronic diseases and geriatric syndromes are influenced by lifestyle factors. The active aging and maintaining functional capacity facilitate health and there are essential in geriatric care.]. At paragraph Methods we added that the study is a mixed design (line 12), as well as the place of recruitment for the participants at line 14 “[from NIGG “Ana Aslan” Bucharest]”, and at line 19 we recovered “17 subjects” and we wrote [10 subjects from NIGG “Ana Aslan” and 7 subjects from GNSPY]”.

C2.3. Keywords: I advise adding: Mixed methods.  I also recommend adding: Psychogerontology, and Mental Health.

R2.3. At Keywords we added [mixed methods]” at line 26.

Comments 3: (Section 1. Introduction

Response 3: We agree with these comments at section 1. Introduction.

C3.1. I suggest removing the 2 subtitles (Context, and Purpose and Objectives), which will allow for a more fluid reading.

R3.1. We removed the 2 subtitles on page 1 (line 29) and 3 (line 123).

C3.2. (1.1. Context)

·                                 I suggest adding a first stanza that contains:

  1. a) demographic background, in relation to the proportion of older people: worldwide, European and/or from the country of origin of the research.
  2. b) epidemiological background, which supports the practice of Yoga in older people in general: Risk Factors, pathologies, or Geriatric Syndromes.

R3.2. On page 1 we added at first part of Introduction the paragraphs reflecting the demographic background and epidemiological background “[Demographic change is an important and current issue… are common problems in older adults]” and we added references 1-15. In addition to this part, we changed the paragraph from page 2 (line 83-84): “Aging is a lifelong process and… many chronic diseases and geriatric syndromes are influenced by lifestyle factors.” and reference [5] became [16].

C3.3. It is necessary to define what Yoga is, and the types of Yoga that exist. At the same time, which ones are most frequently used by older people, and reported in the literature.

      It should be indicated in which situations Yoga is used/indicated (as an intervention) in older people, its effects (reported by other research) on biological, functional, psychological, social, and spiritual health. Then, special emphasis should be placed on the inclusion and social participation of older people, which will later be the emphasis of the profile to be found in this research.

      Similarly, indicate whether Yoga is contraindicated in any health condition in older people.

      All the above, before delving into the more specific aspects of psychogerontology in relation to Yoga in older people.

      Also, the characteristics of Yoga (as an intervention) should be studied for it to have effects on the mental health of older people, in terms of the minimum requirements (parameters): intensity, duration of a session, weekly frequency of sessions, duration of the intervention in months. To do so, I recommend searching for/referencing a systematic review and/or meta-analysis.

R3.3. We have integrated paragraph “[Yoga is a product of the civilization of India…]” and reference [17] for definition of Yoga (line 35). We added paragraph “[There are nine primary styles commonly practiced…]” and references [18-22] for the types of Yoga (line 38). Reference [1] became [23].

On page 2 we added paragraph “[Yoga practices can be adapted for the specific needs of the older adults…]” and references [24-25] for relation of Yoga with older people (line 43). References [2-3] became [26-27]. In addition to this part about Yoga, we changed the paragraph from page 2 (line 70-77): “Yoga, as science and as art of a conscious life… anxiety, stress and depression.” and reference [4] became [28].

We have not developed the informations regarding the parameters of yoga sessions because this was not our objective for this study. We will develop these infromations in a further article.

C3.4. These last aspects (parameters) can also be added when describing the Erasmus+ Project, allowing for a better understanding of the ongoing project and the Yoga intervention with older people.

             Also, it should be explained what adaptations are made to Yoga with older people and that       we  re addressed in their project.

    When describing the Erasmus+ Project, the objectives are mentioned. This creates some       confusion with the objective of this research. Please review this aspect so as not to confuse the  reader.

R3.4. We have integrated the above information within the Introduction, to the extent of the aim of the present paper. More in-depth information on each of the advised levels will be included in a future paper, namely the main paper of the Erasmus project as they do not fit the objectives of this first study. On page 2 for a better understanding of the objective of this research we removed the paragraphs “Our project’s main objective… older people with reduced possibilities.” (line 78-90) and before last paragraph “An optimal beneficiary profile refers to…” (line 91) we added paragraph “[To develop an effective Yoga intervention program…  Thus, the idea of creating an optimal beneficiary profile emerged.]”. On page 3 references [6-8] became [29-31]

C3.5. Line 119-120. “Numerous studies” are indicated, but there are no bibliographical references to support this.

R3.5. We have corrected this. On page 3 (line 119-120) we removed “Myriad studies and design research… profile created” and we wrote “[we created a profile of an optimal beneficiary through…]”  

C3.6. (1.2. Purpose and Objectives) The aim of the research is the last stanza. The rest is introduction.

R3.6. On page 3 we have left the aim of the research as the final part of the introduction. We changed paragraph “It was also particularly useful in helping us…” (line 126-128) before last paragraph and we removed paragraph “The idea of identifying a profile…“ (line 129-130). After last paragraph (line 140) we added “[We wanted to communicate what the potential user experience could be…]”

Comments 4: (Section 2. Methods)

Response 4: We agree with these comments at section 2. Methods

C4.1. The title for this section should be: Materials and Methods

R4.1. We removed the title (line 141) for this section “2. Methods” and we wrote “[2. Materials and Methods]”.

C4.2. It should be indicated type and design of the research. Therefore, I recommend specifying that it is a “mixed methods” research, and specifically indicating the mixed research design used (with its respective bibliographical reference).

R4.2. On page 4 we removed the title (line 142) for the subsection “2.1. Participants” and we wrote “[2.1. Study Design and Participants]”. At first part of this subsection, we added the paragraphs indicating the mixed research design used “[This study is based on a mixed-methods research design…]”.

C4.3. Then, in a stanza, specify the research design for the quantitative part, and then for the qualitative part. With its respective bibliographic reference for each design approach.

The two previous points will give a better understanding and replicability of the research.

I recommend using and reporting, as a supplementary table, some checklist for qualitative research (COREQ-32 or similar) of this research.

R4.3. At second part of the subsection “[2.1. Study Design and Participants]” we added the paragraphs specifying the research design for the quantitative part, and then for the qualitative part “[The study included two groups…]”

C4.4. The ethical aspects of all this research may be better placed in this section.

R4.4. Agreed. The ethical aspects of all this research was placed in this section. On page 4 we added subsection “[2.2. Ethical Considerations]” and paragraph “[The study was conducted in accordance with the ethical principles…]”

Comments 5: (Subsection 2.1. Participants)

Response 5: On page 4 we removed the title (line 142) for the subsection “2.1. Participants” and we wrote “[2.1. Study Design and Participants]”.

C5.1. I advise keeping the mixed design structure.

R5.1. At first part of this subsection, we added the paragraphs indicating the mixed research design used “[This study is based on a mixed-methods research design…]”.

C5.2. Therefore, it should be specified that the sampling was not probabilistic.

R5.2. At second part of this subsection we added the paragraphs specifying that the sampling was not probabilistic “[Sampling was non-probabilistic, with participants recruited from…]”

C5.3. Also, it must be specified whether the older people in the group in which the focus groups were conducted (qualitative part) are a subgroup or not, of the group of 50 older people in the hospital in whom the evaluation scales were applied (quantitative part).

R5.3. Also, at second part of the subsection we added the paragraphs specifying the qualitative group “[four focus-groups including 17 participants…]”

C5.4. The sociodemographic characteristics that would be collected from the group of 50 people and from the group of 17 older people should be indicated separately. I also understand that in the 17 people there are 2 additional subgroups that were compared. This should be clarified.

         In each group (50 and 17), the cognitive, functional and educational level of the participants must be specified. Also, indicate whether they used any type of technical aid (cane, walker).

R5.5. On page 4 we added subsection “[2.2. Ethical Considerations]” indicating that “[The study was conducted in accordance with the ethical principles…]”

C5.5. The Ethics Committee that evaluated the project and the approval code must be indicated. Also, if the older people read, understood and signed the informed consent. This for both groups. Check if this is better at the end of 2.

R5.5. On page 4 we added subsection “[2.2. Ethical Considerations]” indicating that “[The study was conducted in accordance with the ethical principles…]”

C5.6. Evaluate whether rows 148 to 150 are a better result, specifically in the first part of the sample characterization that I discuss below (3.).

R5.6. Agreed. We relocated (line 143-151) all paragraph to section 3.Results

Comments 6: (Subsection 2.2. Inclusion criteria)

Response 6: Agreed.

C6.1. I think this section should not be a title. Instead, it should be stanzas from the previous section.

R6.1. On page 4 (line 153) we removed the title 2.2.

C6.2. Specify whether “disadvantaged areas” refers to areas with lower socioeconomic or educational status and/or higher crime rates.

R6.2. We added (line 156) “[areas in which the public transport is not facile to access]”

C6.3. Were there exclusion criteria? These must be incorporated.

R6.3. We added after “Inclusion criteria” the paragraph “[Exclusion criteria]”  

Comments 7: (Subsection 2.3. Measures)

Response 7: Agreed.

C7.1. I advise keeping the mixed design structure.

         Therefore, I would specify which assessment and/or evaluation tools were used for each part of the mixed design.

R7.1. On page 4 we added at the first part of this subsection (line 171) the paragraph “[The study employed a mixed-methods approach…]”   

C7.2. For each subscale of the CASE-SF tool, you must specify how the score is obtained and/or how it is analyzed. Also, the minimum and maximum score in each subscale.

R7.2. On page 4 we had this information for WHOQoL-BREF (line 180) and we added for CASE-SF scale.

C7.3. It is not clear whether the “Subjective reports ” were diagnoses that were in the clinical record or were spontaneously reported by the older people. Were these used as a score?. These aspects need to be clarified.

R7.3. On page 5 we added (line 202) this information “[Medical diagnosis were extracted from clinical records…]”  

C7.4. When I get to the “focus groups” part, it is not clear to me whether the 10 people were part of the group of 50 or not. This needs to be clarified. I only understand now from reading the article that the 7 people alone are the ones who did Yoga.

R7.4. On page 5 we clarrified (line 210) this information “[were not part of the quantitative sample…]”  

Comments 8: (Subsection 2.4. Procedure)

Response 8: Agreed.

C8.1. I advise keeping the mixed design structure.

         Therefore, detail them separately in this section. It is important to specify the conditions under which the evaluation tools (quantitative part) were applied, as well as the focus groups.

R8.1. We have deleted the sections that were redundant and we changed the title and division (based on the mixed methods design) of this subsection as advised.

C8.2. Who applied the assessment instruments?, Was it part of the research team?, Was it another person?, What was their profession?, What level of training did they have in applying the tools?. This should be specified.

R8.2. On page 5 we added (line 234) these informations for “[Quantitative Data Collection]”

C8.3. Who applied the focus groups?, Was it part of the research team, was it another person?, What was their profession?, What level of training in applying focus groups did they have?. This should be specified.

R8.3. On page 5 we added (line 235) these informations for “[Qualitative Data Collection]”  

C8.4. Also, it should be specified what the Yoga intervention was like that the group carried out (I understand that they were part of the focus group). This is a key element of this research. Aspects such as (parameters) should be described in detail: type of Yoga used, intensity, session time, weekly frequency of sessions, duration of the intervention in months, adaptations made with the older people. I recommend adding a figure that reflects the Yoga intervention. This allows us to identify what Yoga experience these older people had.

R8.4. “Yoga Intervention” will be included in a future paper, namely the main paper of the Erasmus project as they do not fit the objective of this study.

Comments 9: (Subsection 2.5. Design and Analysis)

Response 9: Agreed. We have deleted the sections that were redundant and we changed the title and division (based on the mixed methods design) of this subsection as advised.

C9.1. It is possible that the first part of this section (rows 237 to 263) corresponds to the previous section, so I recommend relocating it. Also, that the information contained in these rows be placed as a main or supplementary table. And it should be specified as a “focus group question guide…”. Indicating the common questions and for each group. If there were some differentiated questions for those who practice Yoga and others for those who do not, they should also be added/differentiated.

R9.1. We relocated (rows 237 to 263) to subsection 2.4. Procedure and we created Table 1. “Focus Group Question Guide”.

C9.2. Consequently, this section should simply be called “Analysis”, maintaining the same mixed method logic, with its respective description for the quantitative part first, and then for the qualitative part.

R9.2. We removed (line 236) and we wrote only “Analysis.

C9.3. Consequently, in the quantitative part it is necessary to describe whether the normality of the data was analyzed, the statistical tests that were used, the level of statistical significance, and the statistical program used.

R9.3. Regarding the description of data normality and level of statistical significance, these details were included in the results section. The statistical tests that were conducted, as well as the chosen statistical program were added under the quantitative analysis section.

C9.4. For the qualitative part, the focus groups were audio-recorded and transcribed in full and verbatim? Field notes were taken from the focus groups?, The data were coded?, Who analyzed the information?. This should be specified.

          Similarly, it is necessary to reference the author on whom they based themselves (and the steps they followed according to that author), to describe “the pattern of identification Themes and related topics motifs ”.

          Theoretical saturation was reached in the sample of focus groups?. This should be specified.

          Was any program used for qualitative analysis?. This should be specified.

          Consequently, it should be made clearer how we moved from story to text, codes, themes and subthemes.

R9.4. The details regarding the qualitative analysis were added as well. These included the fact that the focus groups were audio-recorded, that CBD and AS conducted the analysis following the method of Braun and Clarke (2021), and that theoretical saturation was achieved. The codes corresponding to each theme, as well as specific quotes from the participants are added in a table, under the Results section. We added reference [38].

Comments 10: (Section 3. Results)

·                                 There are no qualitative results in the results section, they all seem to be in the discussion. It's not clear if those sections have been cut and pasted wrong sequence but as it stands, the manuscript layout is confusing

Response 10: We have made the sequence of steps in results section clearer by respecting the order of quantitative analysis and qualitative analysis.

C10.1 (3.1. Quantitative Analysis)

·                                 As I mentioned before, a more detailed table is missing with the sociodemographic and health characteristics of both groups (50 and 17). And the respective subgroups in the case of the focus groups. Please, to be incorporated.

·                                 It seems that rows 282 to 285 correspond to some section of “ Materials and Methods”. Analysis ?

·                                 It was not used in the model 2 ( Table 2) the variable: “Perceived impairment of one's quality of life.” ?.

R10.1. The quantitative results were edited, with more information on the regression assumptions, regression models and their level of significance being added for clarity. The details pertaining to the sample characteristics were added in the Participants subsection of the Materials and Methods section. Indeed, one’s perceived impairment of one’s quality of life was not introduced in model two, and the error of mentioning it was corrected.

C10.2 (3.2. Qualitative Analysis)

·                                 I recommend that the title of each topic be in italics, as is the case with the “Health” topic.

·                                 I think that somewhere in topic 4, the idea of “personal barriers to doing Yoga” should be outlined.

·                                 Figure 1 may benefit from adding to the figure the “levels” of the Health theme, as well as the “ leitmotifs ” identified in the various themes.

·                                 Textual citations should be associated with the results and not with the discussion.

·                                 Rows 360 to 384 correspond to discussion and should be placed there.

·                                 In rows 385 to 389 it is not clear what profile of older people is being referred to. I recommend carrying out a sort of triangulation between the quantitative and qualitative results, and explaining in the text what profile of older people would benefit most from a yoga intervention of these characteristics. Apparently this is in rows 563 to 584.

·                                 It is also not clear and/or explicit how these variables relate to the inclusion and social participation of older people. This can be discussed, however.

R10.2. Thank you for these useful comments. We have changed the format of the titles of the themes and we made sure to mention the barriers in topic 4. Textual citations were added in a table in the results section and the necessary changes between the two chapters were made. The profile was added based on the triangulation method.

Comments 11 (Section 4. Discussion)

Response 11: We rearranged sections of discussion and conclusions and completed the bibliographic part.

·                                 C11.1. there are significant quotes in the discussion section that should be in results. R11.1. We rearranged sections of results and discussion.

·                                 C11.2. the description of an optimal participant is results and should be moved to that section.

R11.2. We moved to results.

·                                 C11.3. There are no conclusions in the discussion and conclusion section 

R11.2. We rearranged section of conclusions.

4. Response to Comments on the Quality of English Language

Point 1: It is clear that this has been translated to English and requires more attention to grammatical issues and proof reading. I suggest getting editorial advice to increase clarity of expression and ideas.

Response 1: We corrected with more attention. On page 3 we removed ”ofn” and we wrote “[of]” (line 137), we removed ”od” and we wrote “[of]” (line 138). On page 4 we removed ”Added” and we wrote “[In addition]” (line 265). On page 6 we removed ”analysze” and we wrote “[analyze]” (line 265).

5. Additional clarifications

At Acknowledgments (line ) we added “[and Sarva Yoga International partners]”

We added 24 references: [1-15], [17], [18-22], [24-25], [38], therefore on page 14-15 reference [5] became [16], [1] became [23], [2-4] became [26-28], [6-8] became [29-31], [9-11] became [32-34], [12-13] became [35-36], [14] became [37], [15] became [39], [19-25] became [40-46], [16-181] became [47-49].

Thank you for these useful comments that led us to improve the quality of our manuscript.

Reviewer 2 Report

Comments and Suggestions for Authors

Thank you for allowing me to review this manuscript describing a yoga intervention to optimise social participation for older people. Social participation for older people is clearly associated with increased health and wellbeing. I have the following comments.

General comments: 

  • English language editing: there are grammatical errors and translation mismatches, with numerous typos throughout the manuscript. 

Methods

  • there is no in-text description of the ethical approval process for this project. A brief reference to that process would provide immediate clarity to the reader
  • focus group questionnaire could be included in an appendix or additional information available on request. 
  • there is no description of analysis approach or analytical framework. References are needed at the very least
  • the stats are appropriate for the study

Results

  • There are no qualitative results in the results section, they all seem to be in the discussion. It's not clear if those sections have been cut and pasted wrong sequence but as it stands, the manuscript layout is confusing

Discussion

  • there are significant quotes in the discussion section that should be in results. 
  • the description of an optimal participant is results and should be moved to that section
  • There are no conclusions in the discussion and conclusion section 
Comments on the Quality of English Language

It is clear that this has been translated to English and requires more attention to grammatical issues and proof reading. I suggest getting editorial advice to increase clarity of expression and ideas.

Author Response

1. Summary

2. Questions for General Evaluation

Reviewer’s Evaluation

Response and Revisions

Does the introduction provide sufficient background and include all relevant references?

Yes/Can be improved/Must be improved/Not applicable

We developed the introduction by adding demographic and epidemiological information, the definition of yoga and types of yoga, the effects of yoga on physical and mental health, and clarified the purpose of this article. We have included all relevant references and we completed the bibliographic part.

Are all the cited references relevant to the research?

Yes/Can be improved/Must be improved/Not applicable

We have included all relevant references.

Is the research design appropriate?

Yes/Can be improved/Must be improved/Not applicable

Therefore, we have developed the introduction section, we clarified the objectives of this article, we rebuild the structure of methodology section, we added tables, we rearranged sections of results and conclusions and completed the bibliographic part.

Are the methods adequately described?

Yes/Can be improved/Must be improved/Not applicable

3. Point-by-point response 2

Are the results clearly presented?

Yes/Can be improved/Must be improved/Not applicable

3. Point-by-point response 3

Are the conclusions supported by the results?

Yes/Can be improved/Must be improved/Not applicable

3. Point-by-point response 4.3.

Comments 1: General comments: 

English language editing: there are grammatical errors and translation mismatches, with numerous typos throughout the manuscript.

Response 1: Thank you for this comment. We agree with it, and we made the changes regarding the English grammar errors.

Comments 2: Methods:

2.1. there is no in-text description of the ethical approval process for this project. A brief reference to that process would provide immediate clarity to the reader                                                                                   Response 2.1.: Agreed. The ethical aspects of all this research were placed in the section 2. Methods (“[2. Materials and Methods]”). On page 4 we added subsection “[2.2. Ethical Considerations]” and paragraph “[The study was conducted in accordance with the ethical principles…]” and on page 5 we added “[Ethical and Confidentiality Considerations]” in the 2.4. Procedure subsection and the paragraph “[All data collection adhered to strict ethical guidelines...]”                                           

2.2. focus group questionnaire could be included in an appendix or additional information available on request. 

Response 2.2.: On page 6 (line 237) we created “[Table 1. Focus Group Question Guide]”, and place it in the 2.4. Procedure subsection.                                                                                                                                                           2.3. there is no description of analysis approach or analytical framework. References are needed at the very least                                                                                                                                                                        Response 2.3.: Agree. We made the following changes: we have inserted the objective for the quantitative analysis, section “[2. Materials and Methods]”  subsection “[2.5. Analysis]” from the section 3. Results, subsection 3.1. Quantitative Analysis (line 281 to 297). “[Quantitative Analysis]”, the paragraph “[The main objective of the... ]” we summarized the continues variables for the quantitative analysis, and added the paragraph “[Age, level of anxiety, depression... ]” and we specified the statistical program that we chose.

The details regarding the qualitative analysis were added as well, “[Qualitative Analysis]”, the paragraph “[The meeting were audio... ]” . These included the fact that the focus groups were audio-recorded, that two researchers conducted the analysis following the method of Braun and Clarke (2021), and that theoretical saturation was achieved. The codes corresponding to each theme, as well as specific quotes from the participants are added in a table, under the Results section. We added reference [38].

2.4. the stats are appropriate for the study

Response 2.4.: The quantitative results were edited, with more information on the regression assumptions, regression models and their level of significance being added for clarity. We corrected the error of mentioning "one’s perceived impairment of one’s quality of life" that was not introduced in model two.

Comments 3: Results:

·                                 There are no qualitative results in the results section, they all seem to be in the discussion. It's not clear if those sections have been cut and pasted wrong sequence but as it stands, the manuscript layout is confusing

Response 3: Agree. We have made the sequence of steps in results section clearer by respecting the order of quantitative analysis and qualitative analysis. On page 8 we added a table for qualitative results “[Table 4. Example for each theme that ocurred during the focus groups]”  and we place it here, rows from 392 to 562, paragraph “[The extent to which the goals of the older person... ]” which was in the section 4. Discussion and Conclusion,  4.1. Structure subsection.

Comments 4: Discussion

Response 4: Agree. We have, accordingly made the necessary changes.

4.1. there are significant quotes in the discussion section that should be in results.                             

Response 4.1. We rearranged sections of results and discussion

4.2. the description of an optimal participant is results and should be moved to that section

Response 4.2. We moved to results.

4.3. There are no conclusions in the discussion and conclusion section 

Response 4.3. We rearranged section of conclusions.

4. Response to Comments on the Quality of English Language

Point 1: It is clear that this has been translated to English and requires more attention to grammatical issues and proof reading. I suggest getting editorial advice to increase clarity of expression and ideas.

Response 1: We corrected with more attention. On page 3 we removed” ofn” and we wrote “[of]” (line 137), we removed ”od” and we wrote “[of]” (line 138). On page 4 we removed ”Added” and we wrote “[In addition]” (line 265). On page 6 we removed” analysze” and we wrote “[analyze]” (line 265).

5. Additional clarifications

At Acknowledgments (line 600 ) we added “[and Sarva Yoga International partners]”

We added 24 references: [1-15], [17], [18-22], [24-25], [38], therefore on page 14-15 reference [5] became [16], [1] became [23], [2-4] became [26-28], [6-8] became [29-31], [9-11] became [32-34], [12-13] became [35-36], [14] became [37], [15] became [39], [19-25] became [40-46], [16-181] became [47-49].

Thank you for these useful comments.

Round 2

Reviewer 1 Report

Comments and Suggestions for Authors

Dear authors, 

I hope you are well.

I appreciate having addressed most of the suggestions and/or questions I raised, as well as providing the corresponding explanations.

I see substantial improvements in your article.

I wish you much success in this and other projects.

Best regards!